# Hydroxylamine-mediated C–C amination via an aza-hock rearrangement

Tao Wang[1], Philipp M. Stein[1], Hongwei Shi[1], Chao Hu[1], Matthias Rudolph[1] & A. Stephen K. Hashmi [1,2✉]

Despite the widespread use of anilines, synthetic challenges to these targets still exist. Selectivity is often an issue, when using the traditional nitration-reduction sequence or more modern approaches, including arene C–H aminations catalyzed by transition metals, photo-sensitizers, or electrodes. Accordingly, there is still a need for general methods to rapidly, directly access specific isomers of substituted anilines. Here, we report a simple route towards the synthesis of such motifs starting from benzyl alcohols, which are converted to anilines by the use of arylsulfonyl hydroxylamines, via an aza-Hock rearrangement. Good to excellent yields are observed. The method is applicable to various benzyl alcohol surrogates (such as ethers, esters, and halides) as well as simple alkylarenes. Functionalizations of pharmaceutically relevant structures are feasible under the reaction conditions. Over ten amination reagents can be used, which facilitates the rapid assembly of a vast set of compounds.

[1] Organisch-Chemisches Institut, Heidelberg University, Im Neuenheimer Feld 270, 69120 Heidelberg, Germany. [2] Chemistry Department, Faculty of Science, King Abdulaziz University (KAU), Jeddah 21589, Saudi Arabia. ✉email: hashmi@hashmi.de

Anilines are of paramount importance to all aspects of chemical science (pharmaceuticals, agrochemicals, organic chemicals, and natural bioactive products)[1–6]. Such motifs are typically accessed by a nitration-reduction sequence or by transition metal-catalyzed cross-coupling reactions (Buchwald-Hartwig[7,8], Ullmann[9–11], and Chan-Lam reaction[12–14]). In the case of cross-couplings, pre-functionalized aryl (pseudo) halides are required. To address this drawback, modern methods for direct arene C–H aminations via organometallic chemistry[15,16], photochemistry[17], and electrochemistry[18] were invented. However, specific site-selectivity is a significant challenge for all of the mentioned protocols—*ortho-* versus *para-* selectivity for electron-rich substrates; *ortho-* versus *meta-* versus *para-* selectivity for electron-deficient substrates.

Alternatively, direct C–C amination is a method to address the site-selectivity problem. Many examples for transition metal-catalyzed functionalizations via decarboxylative aminations[19] as well as metal-free reactions (Lossen[20,21], Hoffmann[22], Curtius[23], Schmidt[24] and Neber rearrangements[25]) from carboxylic acids and their derivatives exist, but several synthetic steps are needed. Only a few publications report on direct access to anilines under metal-free conditions in a one-step approach. The Beckmann rearrangement[26,27] is a classical way to afford anilines from ketoximes or surrogates. A pioneering work reported by Walter[28] relied on a similar strategy starting from 2-benzoylpyridine ketoxime or its *O-p*-toluenesulfonate, followed by a Beckmann rearrangement and hydrolysis to provide 2-amino pyridine or anilines, but only a limited number of examples were presented. A recent patent[29] described anilines synthesis using the Beckmann rearrangement of aryl ketoximes, which still needed conc. HCl and high temperature (100 °C) to facilitate the transformation. Later this method was improved by Uchida[30], who realized the direct deacylative amination of acyl arenes with ketoxime benzenesulfonate, affording anilines through a cascade reaction under mild conditions. However, via this methodology only primary anilines are accessible. Still, metal-free direct dealkylative aminations remain an underexplored area, especially for the difficult C(sp²)–C(sp³) bond cleavage (bond dissociation energy: C(sp²)–C(sp³) 98–100 kcal/mol)[31,32]. One other way to address this problem is the Schmidt rearrangement of α-azido ethylbenzene intermediates. An initial intramolecular C–C amination was enabled from benzylic azides in the presence of Lewis acids (CF₃SO₃H or EtAlCl₂) as the catalyst, in which C–C aminated products served as side-products[33]. Later the

Ning group[34,35] showed that alkylarenes or secondary benzyl alcohols were feasible to undergo C–C aminations with DDQ as oxidant and NaN₃ or alkyl azides as the aminating reagents. This is especially remarkable as this paves the way for these types of aminations starting from alcohols or alkylarene substrates instead of the commonly used ketones or carboxylic acids. Later, this technique was developed further by the same group[36], replacing common oxidants with electrochemistry. However, highly toxic and explosive sodium azides or alkyl azides are essentially needed for this Schmidt-like transformation, thus safety precautions should be taken when undertaking such reactions. As a consequence, safe and easy-to-handle protocols are still needed for direct C–C aminations. Our approach was influenced by the synthesis of phenols via the Hock rearrangement, which is the key step for the industrial phenol synthesis (named Cumene-Phenol process or Hock process)[37], the basis for the production of millions of tons of phenol every year (Fig. 1a). In the Hock rearrangement, cumene hydroperoxide, as a key intermediate, is transformed into phenol in acidic solvent after rearrangement and hydrolysis. A related migration of an aryl group in this case onto a nitrogen atom was realized by Falck and Kürti[38], who used reactive ArB(OH)₂NH₂OX intermediates bearing a weak N–O single bond as a pathway to primary anilines. The key intermediates were generated by the nucleophilic attack of hydroxylamine derivatives onto arylboronic acids. Inspired by this technique, we hypothesized that cumene hydroxylamine derivatives, owing to the weak N–O bond (similar to the O–O bond in cumene hydroperoxide), are susceptible to a Hock-type rearrangement in an acidic solvent, yielding anilines as products (so-called aza-Hock rearrangement, Fig. 1b). Evidence that an aza-Hock rearrangement is feasible can be obtained by Hoffmann's report[39], showing that *N*-alkyl-*O*-(arylsulfonyl)hydroxylamines undergo cationic carbon-to-nitrogen rearrangements to form imines, which upon hydrolysis generate anilines (two examples). Here we present a mild, general and scalable, chemoselective method for C–C aminations. It has a broad scope, including secondary benzyl alcohols, simple alkylarenes in combination with arylsulfonyl hydroxylamines (ArSO₂ONHR) as aminating reagents, delivering primary or secondary anilines in good to excellent yields (>70 examples, Fig. 1c).

## Results

### Reaction design and optimization towards secondary anilines.
Hydroxylamines and their derivatives are powerful aminating reagents, which are often used for arene C–H and X-H

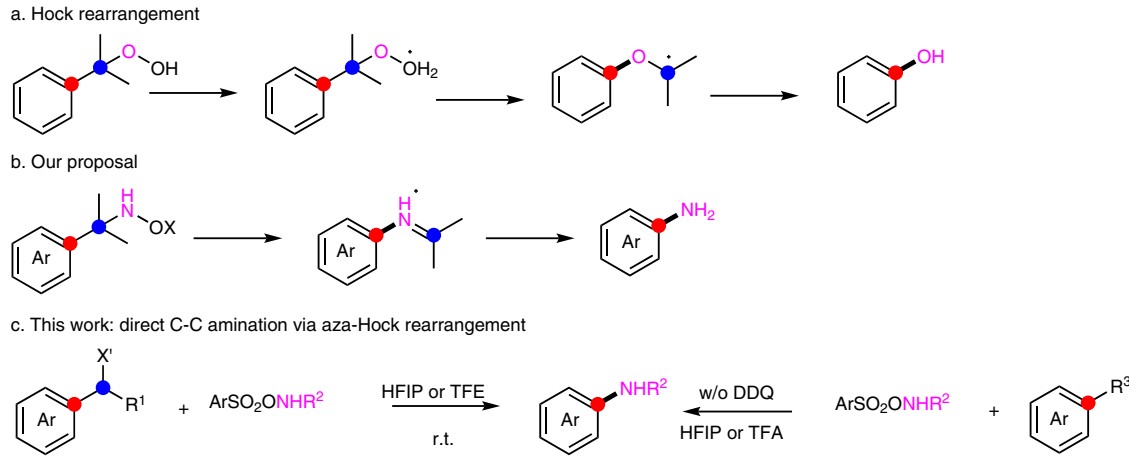

**Fig. 1 Hock rearrangement and reaction design for C–C amination.** Color labels are used to visualize the disconnection. **a** Hock rearrangement for phenol synthesis. **b** Our proposal for aniline synthesis. **c** This work with direct C–C amination via aza-Hock rearrangement. *DDQ* 2,3-dichloro-5,6-dicyano-1,4-benzoquinone, *HFIP* hexafluoroisopropanol, *TFE* trifluoroethanol, *TFA* trifluoroacetic acid.

**Table 1 Optimization of the C–C amination towards secondary anilines.**

| Entry | Solvent | Yield[a] |
|---|---|---|
| 1[b] | DCM (0.4 M) | Trace |
| 2[b] | CH$_3$CN (0.4 M) | Trace |
| 3[b] | MeOH (0.4 M) | Trace |
| 4[b] | TFE (0.4 M) | 60% |
| 5[b] | HFIP (0.4 M) | 72% |
| 6 | HFIP (0.4 M) | 75% |
| 7[c] | HFIP (0.4 M) | 77% |
| **8** | **HFIP (0.2 M)** | **78% (74%[d])** |
| 9 | HFIP (0.1 M) | 81% |
| 10[e] | HFIP (0.2 M) | 78% |
| 11[f] | HFIP (0.2 M) | 78% |
| 12[g] | HFIP (0.2 M) | 16% |
| 13[h] | HFIP (0.2 M) | 60% |

Reaction conditions: alcohol (0.2 mmol), **2a** (0.22 mmol), solvent (1 mL), 12 h, r.t.
The bold line documents the optimized conditions.
[a]NMR yield (1,3,5-trimethoxybenzene as internal standard).
[b]**2a** (0.20 mmol) was used.
[c]**2a** (0.24 mmol) was used.
[d]Isolated yield.
[e]Under nitrogen atmosphere.
[f]Under dark condition.
[g]TsOH•H$_2$O (0.22 mmol) as additive.
[h]TFA (0.22 mmol) as additive.

aminations (X = O, N, S, P)[40–45] as well as Schmidt-type reaction[46], serving as alternative ways to introduce amino groups on various chemical skeletons. Recently, our group[47] published a metal-free arene C–H amination using ArSO$_2$ONHR in HFIP. All this prompted us to investigate the possibility of harnessing hydroxylamine derivatives for direct arene dealkylative aminations. We supposed that 2-phenylethanol, as an electrophilic precursor in fluorinated alcoholic solvents, could react with nucleophilic TsONHMe to form the reactive Hock-type intermediate, which triggered by the weak N–O bond in situ might undergo an aza-Hock rearrangement, which after hydrolysis of an intermediate iminium ion would give valuable anilines. Already the initial exploratory experiment using 1-(4-methoxyphenyl)ethan-1-ol **1a** and TsONHMe **2a** led to aniline **3a** in a good yield (Table 1, entry 6). The aminating reagent **2a** (0.22 mmol) in combination with 0.2 mol/L **1a** in HFIP turned out to be the optimal reaction condition and 74% isolated yield was obtained after the systematic screening. The choice of solvent was crucial: nonfluorinated polar or nonpolar solvents were not effective, but fluorinated alcohols worked well and HFIP proved to be superior to TFE (entries 1–5). A larger excess of the aminating reagent and higher concentrations slightly increased the yield (entries 5–7 and entries 6, 8, 9). A nitrogen atmosphere or exclusion of light did not affect the C–C amination (entry 10, 11) and the addition of p-toluenesulfonic acid or TFA suppressed the formation of anilines due to the competing water elimination from the alcohol (entry 12, 13).

**Substrate scope in the synthesis of secondary anilines**. The scope of this dealkylative amination is striking and almost all tested secondary benzyl alcohol partners were transformed in excellent chemoselectivity. Alcohols that can be used in the C–N bond formation are exemplified in Fig. 2. Commercially available secondary benzyl alcohols (**1a–1d**) proceeded smoothly with the protocol. Higher yields were obtained when electron-donating groups were

installed on the arene core. Tertiary alcohols (**1e**, **1f**) also reacted with **2a**, providing excellent yields. Notably, the electron-rich 4-(methylamino)phenol **3g**, which is sensitive to oxidants, was successfully synthesized with the aid of the related aminating reagent MsONHMe **2b**, which was first used in C–H amination by our group[47]. Alcohols (**1h–1j**) bearing electron-withdrawing groups (Br, Cl, OTs) were also compatible. In the case of strong electron-withdrawing groups (CN, **1k**; NO$_2$, **1l**), an additional methoxyl group on the benzene core was essential to regulate the electronic properties. 1-(3-(Methylamino)phenyl)ethan-1-ol **3m** was the only product when bifunctional 1,1'-(1,3-phenylene)bis(ethan-1-ol) **1m** was implemented in the reaction, which we attribute to the iminium salt for aniline **3m** in HFIP (See discussion). Di- or tri-substituted alcohols (**1n**, **1o**) were also tested and the formation of C–N bonds was achieved in good yields, especially hindered alcohol **1o** was also tolerated. Biphenyl- and diphenyl ether substructures were well tolerated as well as naphthalene (**1p–1r**). It needs to be pointed out that 4-bromo-N-methylbenzene **3s** in combination with 4-bromobenzaldehyde in 70% isolated yield (key side-product for the reaction, see mechanistic discussion below) was accessed when diphenyl methanol **1s** was treated with **2a**. Moreover, heteroarenes and heterocycles (**1t–1x**) were viable for the transformation, even the triple bond of **3v** was kept intact in the presence of oxidative **2a**. Noteworthy, in contrast to diphenyl ether **1q**, the fused benzene ring **1t** delivered a better yield, which is attributed to the participation of lone pair in the aromaticity of the five-member ring.

Given these collective findings, we anticipated that our protocol would be amenable for late-stage functionalizations of natural products, drugs, or chemical waste. Fragrances (tonalide **1y**, coumarin **1z**) was therefore prepared and exposed to the aminating reagent **2a**, providing the products in moderate to good yield; however, for coumarin, a higher concentration of **2a** was needed to achieve the dealkylative amination and inhibit the formation of coumarin ether as byproduct (see SI). Drugs (gemfibrozil **1aa**, naproxen **1ab**, and fenofibric acid **1ac**) were also tested under the

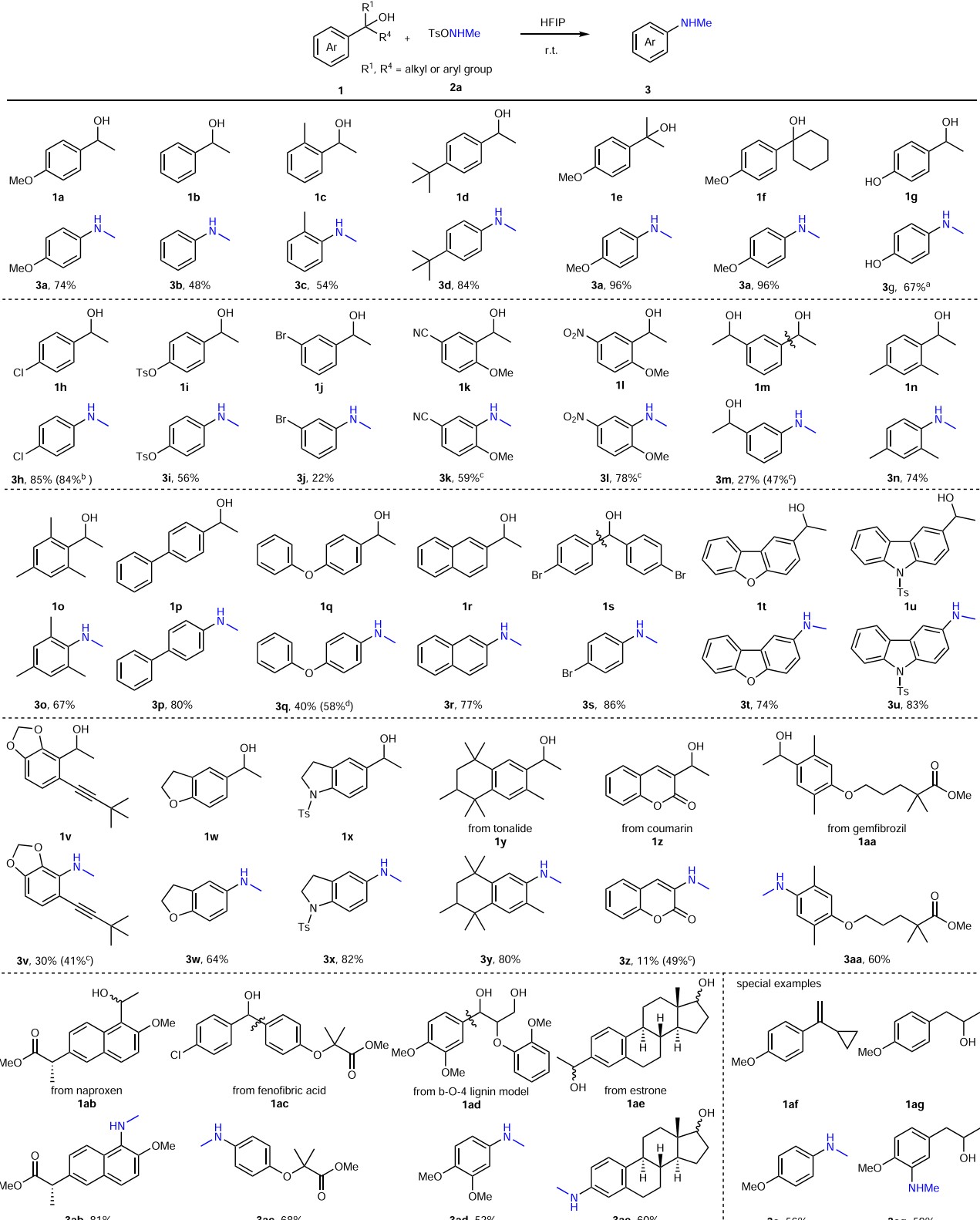

**Fig. 2 Scope with regard to the alcohol in the synthesis of secondary anilines.** Reaction conditions: alcohol (0.2 mmol), **2a** (0.22 mmol), HFIP (hexafluoroisopropanol) (1.0 mL), 12 h, r.t.; [a]MsONHMe (Ms = methanesulfonyl) (**2b**, 0.22 mmol) instead of **2a**; [b]**1h** (4 mmol), **2a** (4.4 mmol), HFIP (20 mL); [c]**2a** (0.44 mmol), HFIP (0.5 mL); [d]**2a** (0.26 mmol) was used; *Ts* toluenesulfonyl.

**Fig. 3 Scope with regard to the hydroxylamines in the synthesis of secondary anilines.** Reaction conditions: alcohol (0.2 mmol), aminating reagents (0.3 mmol), HFIP (hexafluoroisopropanol) (1 mL), 12 h, r.t.; [a]**2b** (0.22 mmol); [b]1-(3,4-dimethoxyphenyl)ethan-1-ol (**1ah**, 0.2 mmol) instead of **1a**; [c]TFE (trifluoroethanol) (1 mL) instead of HFIP; *Ts* toluenesulfonyl, *Ms* methanesulfonyl.

standard conditions, delivering the amination products in good yield; and again, the aldehyde was also isolated from the reaction of **1ac** and **2a** (see SI). Bio-renewable lignin **1ad** also underwent the C–C amination in a good yield, which opens up a window for the synthesis of valuable anilines from waste chemicals. The aminating reagent **2a** could also selectively cleave the C–C bond of benzyl alcohols attached to natural products (estrone, **1ae**), leaving the aliphatic alcohol intact. The robust and practical nature of our strategy was also demonstrated by the large-scale preparation of **3 h** in almost the same efficiency. Furthermore, olefin and other alcohols were also investigated for the C–C amination: α-alkyl substituted styrene, 1-(1-cyclopropylvinyl)-4-methoxybenzene **1af**, was smoothly transformed into aniline **3a** in moderate yield; 1-(4-methoxyphenyl) propan-2-ol **1ag** gave the C–H amination rather than C–C amination product when treated with **2a**, while (4-methoxyphenyl)methanol as primary benzyl alcohol did not react with **2a**.

As a next step, we evaluated the possibility of introducing different N-substituents via the hydroxylamine derivatives. 12 different aminating reagents (**2b** or ArSO₂ONHR, the latter accessed via Mitsunobu reactions) were evaluated for this method. As depicted in Fig. 3, various secondary anilines (**3a**, **4c–4i**) were synthesized from hydroxylamines (**2b–2i**) in excellent yields, even for sensitive aminating reagents (**2c**, **2d**) and sterically hindered reagents (**2e**, **2f**). An alkyl chloride was kept intact during the synthesis of **4i**. A carbamate group **4j** was successfully installed on the benzene core when **1a** was treated with **2j** in less acidic TFE, and a propargyl amine **4k** unit was smoothly introduced to anisole. It is noteworthy that the intermolecular C–C aminations delivered products **4l**, **4m** in excellent yield, no intramolecular aziridination[48] or arene C–H amination[49] side-products were observed despite the reported aziridination of **2l** and the arene C–H amination of **2m** in HFIP

or TFE. A tertiary aniline derivative was not accessible with TsONMe₂ as aminating reagent under these conditions.

**Reaction optimization towards primary anilines.** Next, we wanted to address the even more challenging synthesis of primary anilines via the C–C amination protocol (Fig. 4). First, over 20 aminating reagents were tested. The optimization of the direct conversion to primary aniline from benzyl alcohol is summarized in Table 2. N-Boc arylsulfonyl hydroxylamines (**5a–5g**) or MsONHBoc **5s** delivered the dealkylative amination products in moderate yields (entries 1–7, 19); among all of the tested ArSO₂ONHBoc reagents, the sterically hindered *tert*-butyl ((((2,4,6-triisopropylphenyl)sulfonyl)oxy)carbamate **5e** demonstrated the best performance, affording a moderate yield (50%) in HFIP. However, O-benzoyl or pivaloyl hydroxylamines (**5h–5j**, **5r**) only led to minor or trace amounts of product (entries 8–10, 18). Besides, other commercially available reagents like (2,4-dinitrophenyl)hydroxylamine (DNPHA, **5k**) and O-(diphenyl-phosphinyl)hydroxylamine **5l** with a free amino group were less effective for the amination (entry 11, 12). Unprotected O-sulfonyl hydroxylamines turned out to be comparably effective than their corresponding N-Boc derivatives (entry 4 and 13; entry 5 and 14; entry 6 and 15). Noteworthy, O-mesitylenesulfonylhydrox-ylamine (MSH, **5m**) appeared to be the optimal reagent for this conversion in a 59% yield (entry 13, MSH could be stored at −20°C for 1 month with only a little decomposition). Less acidic TFE was confirmed to be a better solvent than HFIP (5% higher yield), which can be attributed to the reduced formation of the elimination product from the alcohol which turned out to be a side pathway (entry 13, 22). Overall, 0.3 mmol of **5m** with

**Fig. 4 Hydroxylamine derivatives used for the synthesis of primary anilines.** *Boc tert*-butyloxycarbonyl, *Piv* pivaloyl, *Ms* methanesulfonyl, *Ts* toluenesulfonyl, *Tf* triflate.

0.2 mol/L alcohol in TFE turned out to be the best conditions for the conversion (79% isolated yield, entry 24). Furthermore, the frequently applied commercially available hydroxylamine-*O*-sulfonic acid (**5q**) delivered only a low yield and TMSONH₂ **5p** was not effective (entry 16, 17). Hydroxylamine triflate salts (**5t**, **5u**) only achieved moderate yields, which we attribute to the formation of free triflic acid which can decompose the starting material (entry 20, 21).

**Substrate scope in the synthesis of primary anilines.** Under the optimized conditions, we next investigated the scope of this methodology (Fig. 5). In a series of benzyl alcohols, with an increasing electron density at the aromatic core, primary anilines were obtained from low to good yields (**6a**, **6d**, **6ai**). Substrates with electron-withdrawing groups (F, **1aj**; NHAc, **1ak**) could be converted with HFIP as the solvent. Di- or tri-substituted alcohols (**1n**, **1al**, **1o**) were amenable for the dealkylative amination in good yields, including sterically hindered substrate **1o**. Besides, biphenyl, naphthalene, and diphenyl ether (**1p–1r**) were tolerated. Not surprisingly, diphenyl methanol **1s** was cleaved by **5m**, forming the desired aniline **6s**. Heterocycles (**1t–1x**) were also competent transformation partners under the reaction conditions as well as pyridine **1am** and thiophene **1an**. The applicability of the methodology for late-stage modifications of drug leads or natural products was demonstrated next. Secondary alcohols derived from fragrances (veratraldehyde **1ah** and tonalide **1y**), drugs (gemfibrozil **1aa**, naproxen **1ab**, and fenofibric acid **1ac**) and a natural product (estrone **1ae**) were successfully converted to anilines, giving good yields. The scalability of the reaction was made evident by the preparation of aniline **6a** on a 4 mmol scale.

**Synthetic applications.** Further applications using our protocols are illustrated in Fig. 6. Our methodology furnished 2-phenyl anilines (**7**, **8**) by a direct C–C amination in just one step. These targets are used as commodity Buchwald ligands precursors (for G2–G4 type ligands). The formal total synthesis of Lidocaine[50] (from **9**) and Chlorambucil[51] (from **10**) could be acquired in very short sequences. Ether and esters (**11**, **12**) (as alternative precursors for benzylic

## Table 2 Optimization of the C–C amination providing primary anilines.

| Entry | Reagent | Solvent | Yield[a] |
|-------|---------|---------|----------|
| 1 | **5a** | HFIP | 28% |
| 2 | **5b** | HFIP | 39% |
| 3 | **5c** | HFIP | 20% |
| 4 | **5d** | HFIP | 48% |
| 5 | **5e** | HFIP | 50% |
| 6 | **5f** | HFIP | 44% |
| 7 | **5g** | HFIP | 48% |
| 8 | **5h** | HFIP | trace |
| 9 | **5i** | HFIP | trace |
| 10 | **5j** | HFIP | 10% |
| 11 | **5k** | HFIP | n.d. |
| 12 | **5l** | HFIP | 16% |
| 13 | **5m** | HFIP | 59% |
| 14 | **5n** | HFIP | 41% |
| 15 | **5o** | HFIP | 33% |
| 16 | **5p** | HFIP | n.d. |
| 17 | **5q** | HFIP | 17% |
| 18 | **5r** | HFIP | n.d. |
| 19 | **5s** | HFIP | 44% |
| 20 | **5t** | HFIP | 48% |
| 21 | **5u** | HFIP | 36% |
| 22 | **5m** | TFE | 63% |
| 23 | **5m** (1.3 eq) | TFE | 72% |
| **24** | **5m (1.5 eq)** | **TFE** | **84% (79%[b])** |
| 25 | **5m** (1.5 eq) | TFE (0.4 M) | 72% |

Reaction conditions: alcohol (0.2 mmol), aminating reagent (0.22 mmol), solvent (1 mL), 12 h, r.t.
[a]NMR yield (1,3,5-trimethoxybenzene as internal standard).
[b]Isolated yield, bold line documents the optimized conditions.

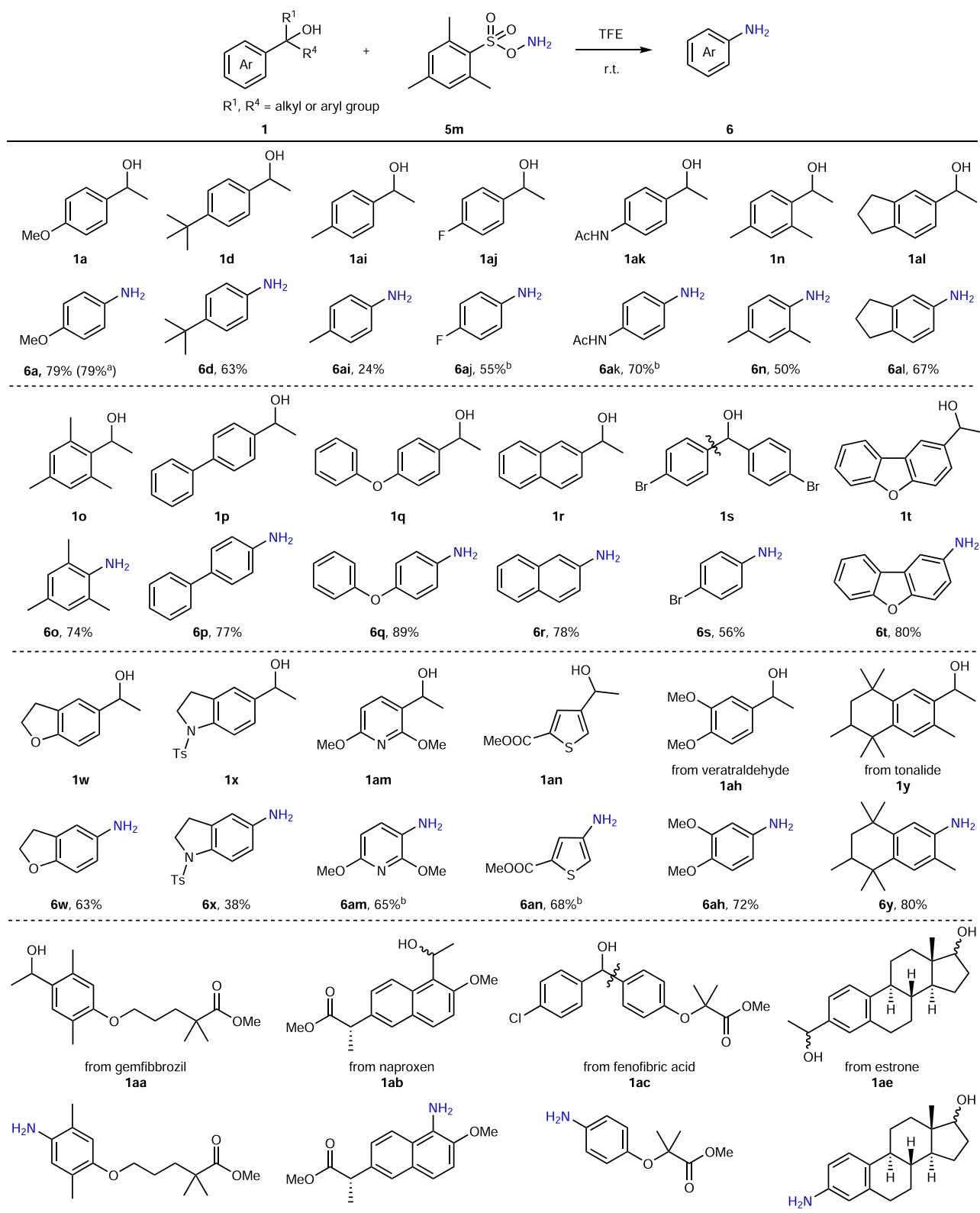

**Fig. 5 Scope with regard to the alcohol in the synthesis of primary anilines.** Reaction conditions: alcohol (0.2 mmol), **5 m** (0.3 mmol), TFE (trifluoroethanol) (1 mL), 12 h, r.t.; [a]**1a** (4 mmol), **5 m** (6 mmol), TFE (20 mL); [b]HFIP (hexafluoroisopropanol) (1 mL) was instead of TFE (1 mL); *Ac* acetyl, *Ts* toluenesulfonyl.

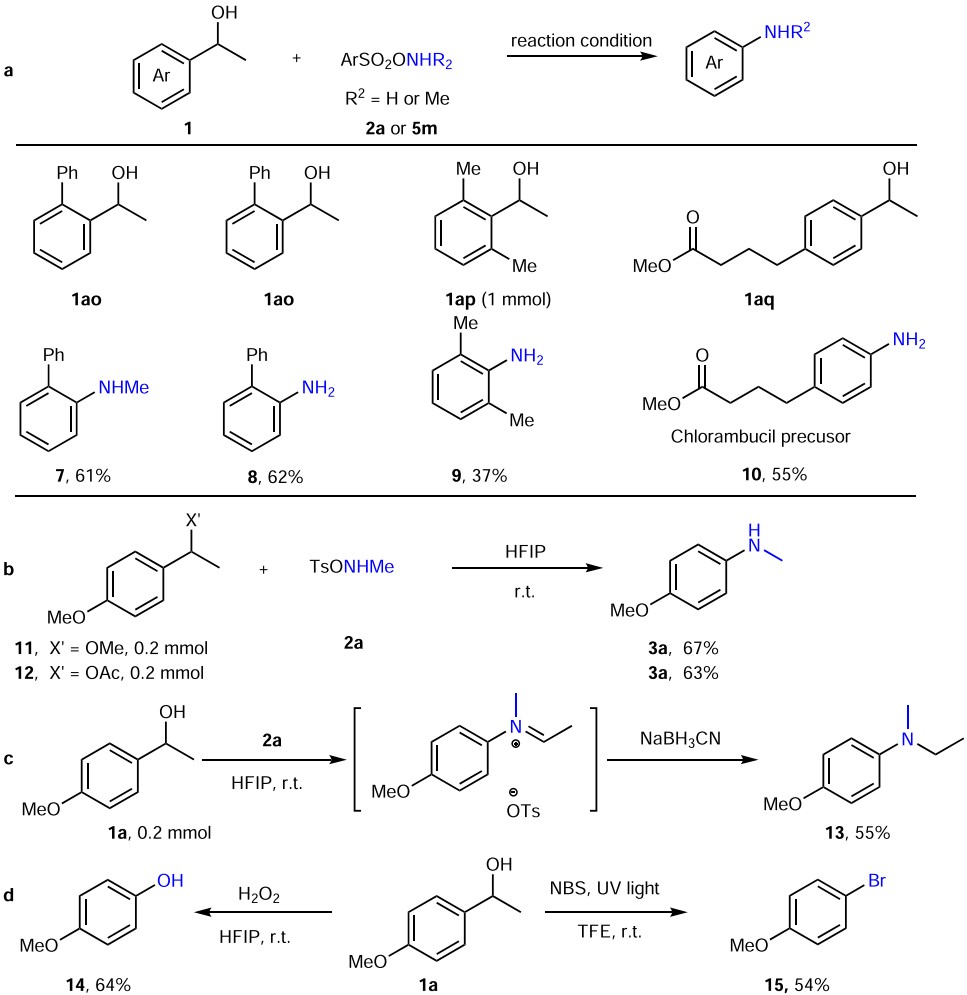

**Fig. 6 Further application of our protocol. a** Four aniline examples for catalysis/bioactive molecules precursors. **b** Benzyl ether/ester as substrate in the aza-Hock rearrangement. **c** Tertiary aniline synthesis with this protocol. **d** Phenol/aryl bromide synthesis with a similar strategy. Reaction conditions: alcohol (0.2 mmol), **2a** (0.22 mmol) or **5 m** (0.3 mmol), HFIP (hexafluoroisopropanol) (1 mL) or TFE (trifluoroethanol) (1 mL), 12 h, r.t.; *Ts* toluenesulfonyl, *NBS* N-bromosuccinimide.

cations) were also useful substrates for the anilines synthesis. Tertiary aniline **13** was affordable by a sequence of a C–C amination of **2a**, followed by a subsequent reduction with NaBH₃CN in one pot. This not only strongly supports the formation of an intermediate imine in the reaction but also further contributes to the synthetic potential of the methodology. Furthermore, under our conditions C–C oxygenations and C–C brominations were also possible. A phenol **14** synthesis from alcohol was feasible when hydrogen peroxide was used instead of a hydroxylamine derivative and a bromobenzene **15** was obtained if NBS under irradiation with UV light was applied.

**Mechanistic investigation**. Concerning the mechanism, we initially considered that a radical pathway might operate related to our previous work[47]. However, all attempts to trap or detect radicals failed (see SI). In order to get more proof, further control experiments were conducted (Fig. 7). Aniline **3h** was also accessible via the bromide derivative of 2-phenylethanol **16** by our protocol. This gives a strong indication that a benzyl cation is the key intermediate of the reaction (the formation of such a cation is a well-known process for benzyl alcohol and halide in HFIP[52–55]). More support for the formation of a cation was provided by the formation of ether **17** from the electron-

deficient alcohol **1i**. Furthermore, a direct dealkylative amination was achieved from 4-isopropyl anisole **18** following a pathway related to Ning's strategy[35]. Besides, 4-bromobenzaldehyde was obtained as a byproduct when diphenyl methanol was treated with **2a**, which proves strong support for the imine hydrolysis. Moreover, a Beckmann rearrangement could be excluded for this reaction, as no aniline was detected when ketone **20** was applied as a substrate under the standard conditions. Accordingly, we propose that the reaction proceeds via an aza-Hock rearrangement in four steps: generation of a benzyl cation via benzyl alcohol solvolysis by HFIP; formation of a reactive O-(1-phenylethyl)hydroxylamine, which gives access to an iminium tosylate salt after aryl migration; and finally the irreversible hydrolysis of the imine yielding the desired aniline after simple workup (Fig. 8). The hydrolysis of protonated imines in the reaction mixture only in the workup with sodium hydrogen carbonate (not in situ with the one equivalent of neutral water formed in the reaction–compare also the selective formation of **13** in Fig. 6, which is based on the same effect), gives an explanation for the selective cleave of one C–C bond instead of two C–C bonds in substrates with two reactive benzylic alcohol groups (**1m** selectively forms **3m**); the iminium group in the intermediate iminium tosylate electronically de-activates the arene ring, the second benzylic alcohol does not react anymore.

**Fig. 7 Mechanistic studies. a** Control experiment with benzyl halide. **b** Side reaction from benzyl alcohol. **c** Control experiment with alkylarene. **d** Byproduct from diphenyl methanol. **e** Exclusion of Beckmann rearrangement for the reaction mechanism. *Ts* toluenesulfonyl, HFIP hexafluoroisopropanol; *TFE* trifluoroethanol.

**Fig. 8 Proposed mechanism of the C–C amination.** *HFIP* hexafluoroisopropanol, *Ts* toluenesulfonyl.

## Discussion

In conclusion, the mild conditions, operational simplicity, and perfect chemoselectivity are the attributes of the C–C amination of benzylic alcohols with $ArSO_2ONHR$ to anilines, which in our opinion is not only attractive for academia but also for industry. As already demonstrated, other precursors for benzylic cations also serve as precursors, which further expands the synthetic utility. In addition, chemoselective C–C brominations and oxygenations are possible under the same conditions. Mechanistically, an aza-Hock rearrangement is proposed by us. Interestingly, despite some early evidence for such a reactivity pattern, until known the synthetic utility of this process was limited and our report might pave the way for further protocols based on this pattern in the future.

## Methods

**General procedure for the synthesis of secondary anilines**. To a solution of the alcohol (0.2 mmol) in 1.0 mL HFIP was added aminating reagent (0.22 mmol) at room temperature under ambient atmosphere unless otherwise stated. The reaction was stirred at room temperature for 12 h (monitored by GCMS or TLC). The reaction was diluted with 1 mL DCM and basified with 1 mL saturated $NaHCO_3$ aqueous solution. The aqueous layer was extracted with DCM (3 mL × 3), and the combined organic layers were washed with 5 mL sat. brine, dried over anhydrous $Na_2SO_4$, filtrated, and concentrated in vacuo. The crude residue was purified by silica gel chromatography with PE/EA (petroleum ether/ethyl acetate) to afford the desired product.

**General procedure for the synthesis of primary anilines**. To a solution of the alcohol (0.2 mmol) in 1.0 mL TFE was added MSH (0.3 mmol) under ambient atmosphere at room temperature unless otherwise stated. The reaction was stirred at room temperature for 12 h (monitored by GCMS or TLC). Then the reaction was diluted with 1 mL DCM and basified with 1 mL saturated $NaHCO_3$ aqueous solution. The aqueous layer was extracted with DCM (3 mL × 3), and the combined organic layers were washed with 5 mL sat. brine, dried over anhydrous $Na_2SO_4$, filtrated, and concentrated in vacuo. The crude residue was purified by silica gel chromatography with PE/EA to afford the desired product.

## Data availability

Authors can confirm that all relevant data are included in the paper and/ or its supplementary information files.

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

## Acknowledgements

T.W., H.S. and C.H. are grateful for Ph.D. fellowships from the China Scholarship Council (CSC). We also thank Maximilian Elter for the technical support; P.M.S. and A.S.K.H. gratefully acknowledge the Hector Fellow Academy for the generous provision of funding.

## Author contributions

T.W. designed, carried out the main part of the experiments, and analyzed data; P.M.S. contributed several experiments to the mechanistic part; All authors participated in discussion; T.W., M.R., and Dr. A.S.K. Hashmi co-wrote the paper; M.R. and A.S.K Hashmi supervised the project.

## Funding

## Competing interests

The authors declare no competing interests.
