## [Peer Review File · Nature Communications]

REVIEWER COMMENTS

Reviewer #1 (Remarks to the Author):

Hashmi and coworkers reported here a method for the preparation of arylamines via C-C bond cleavage from benzyl alcohols. The mild conditions, good compatibility and good yields make this method very practical for synthesis. However, some questions and limitations are obvious in this manuscript, essential improvements should be conducted.

1) TFE and HFIP are very important for the transformation. However, they are too expensive for large-scale application, is it possible to use equivalences or catalytic amount of acids in frequently-used solvents such as DCM and others.

2) Poor tolerance to strong electron-withdrawing groups were shown in Table 2 and Table 5, substrates substituted by other groups such as NO₂ or amide or F were not shown, they are tolerated in previous method with sodium azide as the nitrogen source. Could this limitation be solved by trying more different amination reagents that possess similar reactivity with sodium azide, which is important and meaningful to improve this work.

3) Some relevant work should be cited in the manuscript including Schmidt-type reaction with MeNO₂ as the amination reagent, which generated AcONH₂TfOH as the active species (Science 367, 281–285 (2020)) and C-H amination using hydroxylamine reagents developed by Morandi (ACS Catal. 2016, 6, 8162–8165), Jiao (Chem. Eur. J. 2017, 23, 563 – 567) and Ritter (Chem. Sci., 2019, 10, 2424–2428).

4) Did you detect C-H amination products when using the FeSO₄ as the assistant? if so, how about the yields and selectivity?

Overall, This referee recommend it publication in Nature communication after addressing these questions.

Reviewer #2 (Remarks to the Author):

In the submitted manuscript, Hashmi and coworkers disclose an aza-variant of the widely utilized Hock rearrangement to generate anilines from benzylic alcohols via C-C bond cleavage. While the disclosed aza-Hock rearrangement transformation is mechanistically interesting, it's unclear how useful the reaction would be to the practicing synthetic organic chemist. In the majority of cases, both the amination reagent and the benzylic alcohol starting materials take 1-2 steps to prepare from commercially available reagents. Considering that most of the products prepared in the manuscript can be made directly from commercially available aryl halides and amines using a Buchwald-Hartwig amination, the starting materials for the disclosed aza-Hock rearrangement are often more difficult to make than the products and would likely prevent the widespread adoption of this methodology. Additionally, the substrate scope excludes functionalities that those in the pharmaceutical industry would find relevant, such as heteroarenes (e.g., pyridines, pyrimidines, various azoles) and amides. Due to the perceived lack of applicability of this transformation, I recommend against publishing this manuscript in Nature Communications.

However, I do believe that disclosed transformation is mechanistically interesting. I would encourage the author to pursue additional mechanistic studies to provide more insight into their proposed mechanism and submit a fuller account elsewhere (e.g., Are the reaction kinetics consistent with one of the steps in the proposed mechanism? Would a Hammett plot be able to provide further insight into the nature of the rate determining step? Can the authors gather evidence for the proposed benzylic carbocation and/or iminium ion intermediates through intramolecular carbocationic rearrangements?). They should also remove the "Mechanistically we could prove that an aza-Hock rearrangement operates" line from the conclusion since one cannot prove a mechanism and can instead only gather evidence that supports a proposed mechanism.

Reviewer #3 (Remarks to the Author):

Overall, the novel methodology demonstrated in the manuscript is sound and appropriately referenced and the introduction is useful for bringing a general audience up to speed. The transformation is novel, albeit a logical extension of known methods. This reviewer feels that the substrate Tables 2 and 5 are a bit overwhelming and substrates were not judiciously chosen. These tables could be shortened which would allow for more interesting substrates to be added (see below for comments). The mechanistic studies are scientifically sound.

[Please see the attached document for additional, specific comments.]

I would accept this manuscript for publication to Nature Communications after substantial revisions.

Overall, the novel methodology demonstrated in the manuscript is sound and appropriately referenced and the introduction is useful for bringing a general audience up to speed. The transformation is novel, albeit a logical extension of known methods. This reviewer feels that the substrate Tables 2 and 5 are a bit overwhelming and substrates were not judiciously chosen. These tables could be shortened which would allow for more interesting substrates to be added (see below for comments). The mechanistic studies are scientifically sound.

P1, Introduction, first sentence: no comma after "Anilines"

Scheme 1c: this part of scheme 1 is confusing. The methodology studied is shown in the first arrow (left to right), but the second arrow (right to left) seems to show something else. Is TFA a typo for TFE? I don't actually know what the Ar-R³ is at all, since this paper seems to only show transformations of benzylic alcohols (as depicted in the first left to right arrow)

P3, please introduce alternate reagent MsONHMe before its appearance in the table

Table 2: although it's hinted at, some explicit negative controls would make obvious some of the challenges to this methodology, e.g. 1-(4-nitrophenyl)-ethanol.

Table 2: 1k → 3k (22%) and 1x → 3x (30%): the text suggests that bromides and alkynes are tolerated, but this isn't obvious without explaining the mass balance of these two reactions. Is there remaining starting material or are there byproducts?

Table 2: the yield difference between 1r and 3v is noteworthy but not discussed.

Table 2: to be of broader interest, a more judicious choice of substrates would be useful. There are some substrates that aren't particularly interesting, such as: 1e and 1q (redundant with 1d), similarly 1o, 1s, and 1u don't provide much in terms of scope. Instead, the authors could add more variety to the heterocycle section (specifically 5- and 6-membered rings with multiple heteroatoms are of pharmaceutical interest). The natural products, drugs, and bio-renewable section is a nice addition.

Table 2 / SI: moving 4-methoxybenzyl alcohol (no reaction) to the table would be useful, removing some of the redundant substrates and adding substrates from the SI would also be a better use of space (e.g. styrenes and C-H aminated reaction with 1-(4-methoxyphenyl)propan-2-ol)

Table 5: I'm not sure entry 1aj should be included in the table, since the reaction conditions are substantially different (FeSO₄ additive, previously published in ref. 46). Mentioning it in the text is appropriate

Figure 2: The reduction of the intermediate iminium is a nice touch. However, this reviewer is uncertain if the radical conditions shown with H₂O₂ and NBS are relevant to the transformation being studied.

I would accept this manuscript for publication to Nature Communications after substantial revisions.

REVIEWER COMMENTS

Reviewer #1 (Remarks to the Author):

Hashmi and coworkers reported here a method for the preparation of arylamines via C-C bond cleavage from benzyl alcohols. The mild conditions, good compatibility and good yields make this method very practical for synthesis. However, some questions and limitations are obvious in this manuscript, essential improvements should be conducted.

1) TFE and HFIP are very important for the transformation. However, they are too expensive for large-scale application, is it possible to use equivalences or catalytic amount of acids in frequently-used solvents such as DCM and others.
OUR ACTION: Indeed, TFE and HFIP are expensive than normal solvents, but they can be recycled by simple distillation (Nat. Rev. Chem. 2017, 1, 0088). As you said, 3 equivalents of TFA as acid with DCM as solvent also finish the reaction. But for electron-deficient substrates, HFIP as solvent performed better than DCM with TFA as the acid.

2) Poor tolerance to strong electron-withdrawing groups were shown in Table 2 and Table 5, substrates substituted by other groups such as NO₂ or amide or F were not shown, they are tolerated in previous method with sodium azide as the nitrogen source. Could this limitation be solved by trying more different amination reagents that possess similar reactivity with sodium azide, which is important and meaningful to improve this work.

OUR ACTION: Done, we added example 1l (NO₂) in table 2 and example 1aj (F), 1ak (AcNH) in table 5 in the new revised manuscript.

3) Some relevant work should be cited in the manuscript including Schmidt-type reaction with MeNO₂ as the amination reagent, which generated AcONH₂TfOH as the active species (Science 367, 281–285 (2020)) and C-H amination using hydroxylamine reagents developed by Morandi (ACS Catal. 2016, 6, 8162–8165), Jiao (Chem. Eur. J. 2017, 23, 563 – 567) and Ritter (Chem. Sci., 2019, 10, 2424–2428).

OUR ACTION: Done, we added “⁴⁰⁻⁴⁵ as well as Schmidt-type reaction,⁴⁶” in the new revised manuscript.

4) Did you detect C-H amination products when using the FeSO₄ as the assistant? if so, how about the yields and selectivity?

OUR ACTION: No C-H aminated product was detected with GCMS, TLC and NMR. Other side-product was detected on TLC, but not characterized. As suggested from the third reviewer, we removed this example and added thiophene 1an (with COOMe) in table 5 in the new manuscript.

Overall, This referee recommend it publication in Nature communication after addressing these questions.

Reviewer #2 (Remarks to the Author):

In the submitted manuscript, Hashmi and coworkers disclose an aza-variant of the widely utilized Hock rearrangement to generate anilines from benzylic alcohols via C-C bond cleavage. While the disclosed aza-Hock rearrangement transformation is mechanistically interesting, it's unclear how useful the reaction would be to the practicing synthetic organic chemist. **In the majority of cases, both the amination reagent and the benzylic alcohol starting materials take 1-2 steps to prepare from commercially available reagents. Considering that most of the products prepared in the manuscript can be made directly from commercially available aryl halides and amines using a Buchwald-Hartwig amination, the starting materials for the disclosed aza-Hock rearrangement are often more difficult to make than the products and would likely prevent the widespread adoption of this methodology. Additionally, the substrate scope excludes functionalities that those in the pharmaceutical industry would find relevant, such as heteroarenes (e.g., pyridines, pyrimidines, various azoles) and amides.** Due to the perceived lack of applicability of this transformation, I recommend against publishing this manuscript in Nature Communications.

OUR ACTION:The aza-Hock rearrangement we address here is totally different chemistry from Buchwald-Hartwig amination, which can not simply be compared from the synthetic view; besides, Buchwald-Hartwig reaction needs expensive transition-metals, while our protocol does not. For heteroarenes, we added 1am, 1an in Table 5 in the new revised manuscript.

However, I do believe that disclosed transformation is mechanistically interesting. I would encourage the author to pursue additional mechanistic studies to provide more insight into their proposed mechanism and submit a fuller account elsewhere (e.g., Are the reaction kinetics consistent with one of the steps in the proposed mechanism? Would a Hammett plot be able to provide further insight into the nature of the rate determining step?)

OUR ACTION: Three kinetic plots are shown below: electron-rich substrate reacts faster than electron-neutral and electron-poor substrates. The yields are consistent with isolated yields.

1-phenylethan-1-ol and TsONHMe in HFIP

1-(4-methoxyphenyl)ethan-1-ol, TsONHMe in HFIP

1-(4-chlorophenyl)ethan-1-ol, TsONHMe in HFIP

As you can see in the below spectra, benzyl alcohol was decomposed very quickly after adding TFA in the reaction (measure in the d_2 -DCM with TFA as acid, method shown for the first review). The GC-kinetics with HFIP as the solvent showed the same result as NMR-kinetics: benzyl alcohols was decomposed very fast (No signal of benzyl alcohol around 3 min on GC-MS). This is why it is useless to do Hammett plot (substrates are consumed so quickly). As least, the formation of benzyl cation is not the determined step.

Can the authors gather evidence for the proposed benzylic carbocation and/or iminium ion intermediates through intramolecular carbocationic rearrangements?.

OUR ACTION: two intramolecular benzylic alcohols are tested (shown in the scheme down), but none worked (no intramolecular C-C amination detected), which might be attributed to competitive intermolecular C-C amination.

They should also remove the “Mechanistically we could prove that an aza-Hock rearrangement operates” line from the conclusion since one cannot prove a mechanism and can instead only gather evidence that supports a proposed mechanism.

OUR ACTION: we “Mechanistically we could prove that an aza-Hock rearrangement operates” was instead of “Mechanistically we could prove that an aza-Hock rearrangement operates.”

Reviewer #3 (Remarks to the Author):

Overall, the novel methodology demonstrated in the manuscript is sound and appropriately referenced and the introduction is useful for bringing a general audience up to speed. The transformation is novel, albeit a logical extension of known methods. This reviewer feels that the substrate Tables 2 and 5 are a bit overwhelming and substrates were not judiciously chosen. These tables could be shortened which would allow for more interesting substrates to be added (see below for comments). The mechanistic studies are scientifically sound.

[Please see the attached document for additional, specific comments.]

Text from attached document:

Overall, the novel methodology demonstrated in the manuscript is sound and appropriately referenced and the introduction is useful for bringing a general audience up to speed. The transformation is novel, albeit a logical extension of known methods. This reviewer feels that the substrate Tables 2 and 5 are a bit overwhelming and substrates were not judiciously chosen. These tables could be shortened which would allow for more interesting substrates to be added (see below for comments). The mechanistic studies are scientifically sound.

P1, Introduction, first sentence: no comma after “Anilines”

OUR ACTION: Done

Scheme 1c: this part of scheme 1 is confusing. The methodology studied is shown in the first arrow (left to right), but the second arrow (right to left) seems to show something else. Is TFA a typo for TFE? I don't actually know what the Ar-R3 is at all, since this paper seems to only show transformations of benzylic alcohols (as depicted in the first left to right arrow)

OUR ACTION: NO, it's not a typo, TFA exactly. In scheme 1, transformations of benzyl alcohols dominate the reaction, while we still have other examples (benzyl ether 11 /ester 12 /halide 16, isopropylanisole 18 as well, see Figure 2 and Scheme 2). Ar-R³ means 4-isopropylanisole 18 (see scheme below); in this case TFA gave more C-C aminated product, while C-H aminated product dominated in HFIP

c. This work: direct C-C amination via aza-Hock rearrangement

P3, please introduce alternate reagent MsONHMe before its appearance in the table

Table 2: although it's hinted at, some explicit negative controls would make obvious some of the challenges to this methodology, e.g. 1-(4-nitrophenyl)-ethanol.

Table 2: 1k □ 3k (22%) and 1x □ 3x (30%): the text suggests that bromides and alkynes are tolerated, but this isn't obvious without explaining the mass balance of these two reactions. Is there remaining starting material or are there byproducts?

OUR ACTION: we added “, which was first used in C-H amination by our group” after MsONHMe.

We added example 1l (NO₂) in the new revised manuscript

No other basic side-products were detected with GC-MS and TLC for substrate 1j and 1v in the new revised manuscript. Maybe some acidic side-products were formed from those two substrates (reaction workup with NaHCO₃ aq.)

Table 2: the yield difference between 1r and 3v is noteworthy but not discussed.

OUR ACTION: Done, we added “Noteworthy, in contrast to diphenyl ether 1q, the fused benzene ring 1t delivered a better yield, which is attributed to the participation of lone pair in the aromaticity of the five-member ring.”

Table 2: to be of broader interest, a more judicious choice of substrates would be useful. There are some substrates that aren't particularly interesting, such as: 1e and 1q (redundant with 1d), similarly 1o, 1s, and 1u don't provide much in terms of scope. Instead, the authors could add more variety to the heterocycle section (specifically 5- and 6-membered rings with multiple heteroatoms are of pharmaceutical interest). The natural products, drugs, and bio-renewable section is a nice addition.

OUR ACTION: Done, some redundant examples were removed (eg. 1e and 3e, 1o and 3o, 1u and 3u from table 2; 1u and 6u from table 5) in the old manuscript. Also we added heterocycles 1am, 1an in Table 5 in the new revised manuscript.

Table 2 / SI: moving 4-methoxybenzyl alcohol (no reaction) to the table would be useful, removing some of the redundant substrates and adding substrates from the SI would also be a better use of space (e.g. styrenes and C-H aminated reaction with 1-(4-methoxyphenyl)propan-2-ol)

OUR ACTION: Done. As answered in the above question, redundant substrates were removed and styrene 1af and C-H amination 1ag were added in the new revised manuscript

Table 5: I'm not sure entry 1aj should be included in the table, since the reaction conditions are substantially different (FeSO₄ additive, previously published in ref. 46). Mentioning it in the text is appropriate

OUR ACTION: I'm afraid you didn't get it. FeSO₄, previously published in ref. 46 in the old manuscript, was used for arene C-H amination. Without iron as a additive, the reaction only gave around 20% NMR yield; FeSO₄ could increase the yield a little bit (for the iron role, we could not give a better explanation). And this is the first time of iron used in aza-Hock rearrangement. Now in the revised manuscript, we deleted 1aj in the old manuscript and added 1an (with COOMe) in table 5 in the new manuscript.

Figure 2: The reduction of the intermediate iminium is a nice touch. However, this reviewer is uncertain if the radical conditions shown with H₂O₂ and NBS are relevant to the transformation being studied.

OUR ACTION: for H₂O₂, not sure it's radical pathway or benzyl cation? For NBS, radical pathway is possible as NBS can be initiated by UV and then homolyzed.

I would accept this manuscript for publication to Nature Communications after substantial revisions.

I would accept this manuscript for publication to Nature Communications after substantial revisions.

REVIEWERS' COMMENTS

Reviewer #1 (Remarks to the Author):

My questions have been addressed by the authors. The improved version of the manuscript is acceptable for publication in Nature Communication.

Reviewer #2 (Remarks to the Author):

While I appreciate the authors' attempts to address reviewer critique, I disagree that their methodology cannot be compared to the Buchwald-Hartwig amination. Yes, the two transformations go through distinct mechanisms. However, the fact that the two reactions can be used to make the exact same products warrants comparison when it comes to utility to a practicing organic chemist. The cost of transition metals is not an issue when catalyst loadings are <1 mol%, as they frequently are with Buchwald-Hartwig aminations. I stand with my original recommendation that this manuscript be published elsewhere due to lack of utility.

Reviewer #3 (Remarks to the Author):

accept after revisions